# Robotic Control via Embodied Chain-of-Thought Reasoning

**Michał Zawalski**[*1,2], **William Chen**[*1], **Karl Pertsch**[1,3]
**Oier Mees**[1], **Chelsea Finn**[3], **Sergey Levine**[1]
[1]UC Berkeley, [2]University of Warsaw, [3]Stanford University

**Abstract:** A key limitation of learned robot control policies is their inability to generalize outside their training data. Recent works on vision-language-action models (VLAs) have shown that the use of large, internet pre-trained vision-language models as the backbone of learned robot policies can substantially improve their robustness and generalization ability. Yet, one of the most exciting capabilities of large vision-language models in other domains is their ability to reason iteratively through complex problems. Can that same capability be brought into robotics to allow policies to improve performance by reasoning about a given task before acting? Naive use of "chain-of-thought" (CoT) style prompting is significantly less effective with standard VLAs because of the relatively simple training examples that are available to them. Additionally, the purely-semantic reasoning about sub-tasks common to regular CoT is insufficient for robot policies that need to ground their reasoning in sensory observations and the robot state. To this end, we introduce Embodied Chain-of-Thought Reasoning (ECoT) for VLAs, in which we train VLAs to perform multiple steps of reasoning about plans, sub-tasks, motions, and visually grounded features like object bounding boxes and end effector positions, before predicting the robot action. We design a scalable pipeline for generating synthetic training data for ECoT on large robot datasets. We demonstrate that ECoT increases the absolute success rate of OpenVLA, the current strongest open-source VLA policy, by 28% across challenging generalization tasks without any additional robot training data. Additionally, ECoT makes it easier for humans to interpret a policy's failures and correct its behavior interactively using natural language. Finally, we show that our model learns to transfer ECoT reasonings to unseen embodiments and tasks.

**Keywords:** Vision-Language-Action Models, Embodied Chain-of-Thought Reasoning

## 1 Introduction

Robotic policies trained end-to-end for outputting raw low-level actions in response to observations provide a powerful and appealing learning-based approach to robotic control, obviating the need for complex sensing and control stacks [1, 2, 3]. However, this kind of "reactive" control is not necessarily the best choice in settings that demand broad generalization, such as novel scenes or interactions with unfamiliar objects. In such situations, a person might need to think more carefully: if they were asked to put fruit on one plate and vegetables on another, they might first try to figure out which objects are fruits and which are vegetables, rather than simply performing a learned skill from "muscle memory." Likewise, we would like our robotic policies to both perform well-practiced end-to-end control and to "reason through" novel situations before grounding their commands into actions. Such reasoning might include identifying and locating task-relevant objects, producing a plan to accomplish a task, and translating sub-tasks and observations into movements.

Vision-language-action models (VLAs) – pre-trained vision-language models (VLMs) fine-tuned to produce robot actions – have gained popularity as an approach for leveraging the diversity of Internet data captured within large foundation models [4] in a simple and scalable policy learning

---

*Equal contribution. Website at https://embodied-cot.github.io
Correspondence: m.zawalski@uw.edu.pl, verityw@berkeley.edu, pertsch@berkeley.edu

recipe. Despite achieving state-of-the-art performance across a wide range of tasks and robot embodiments [5, 6, 7], VLAs typically learn a direct mapping from observations to actions without any intermediate reasoning. However, there have been many recent works exploring how language models (which serve as the backbone of VLAs) can be prompted to textually "think step-by-step" about a given task. Such chain-of-thought reasoning (CoT) [8] significantly improves their performance on complex reasoning tasks and is now de-facto a standard practice in language modeling [9].

We thus hypothesize that we can similarly boost VLA performance by training them to *textually reason* about their plan, environment, and motions, thereby allowing them to produce more accurate and robust robot actions. However, simply applying the CoT techniques from language to the robotics domain has several challenges. For one, current VLAs build on relatively small, open-source VLMs that cannot match closed models' ability to perform meaningful reasoning when simply prompted to think step-by-step [8]. Additionally, the most common LLM CoT reasoning task, breaking tasks into sub-tasks, is helpful but insufficient for reasoning about robotic tasks. The VLA policy needs to ground its plans and reasoning in its *observations* of the environment and robot state. Only then can the reasoning direct the agent's attention toward fine-grained spatial or semantic perceptual features that are key to robot manipulation tasks. Put simply, we need VLAs to not only "think carefully", but also "look carefully."

To this end, we introduce **Embodied Chain-of-Thought Reasoning (ECoT)** for VLA policies. In contrast to prior VLAs, embodied chain-of-thought policies perform multiple steps of textual reasoning before predicting the next robot action (see Fig. 1, right). In contrast to existing

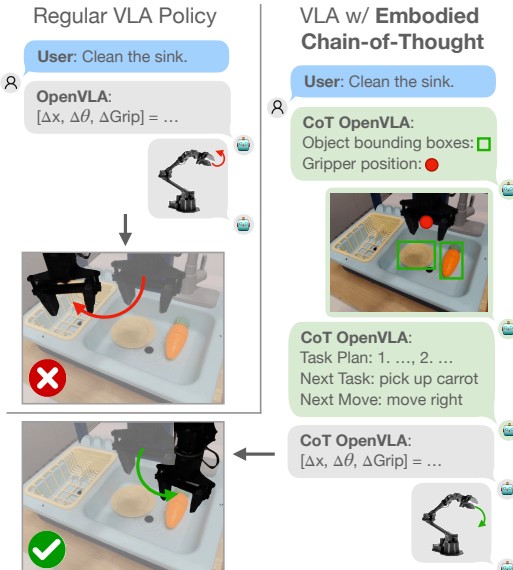

Figure 1: We propose embodied chain-of-thought reasoning for vision-language-action models (VLAs): prior VLAs directly predict the next robot action given the task (**left**), we instead train VLA policies to "think step-by-step" (**right**). Crucially, reasoning through low-level embodied features like object bounding boxes and gripper positions in addition to purely textual CoT elements like sub-task plans, forces the policy to "think carefully" *and* "look carefully" before acting. This approach increases the absolute success rate of state-of-the-art OpenVLA policies [7] by 28% in challenging generalization tasks.

CoT reasoning approaches for LLMs, they interleave semantic-level reasoning about sub-tasks with "embodied" reasoning tasks that require the policy to pay attention to its multi-modal inputs, from predicting bounding boxes of objects in the scene to reasoning about low-level movement primitives that need to be executed based on the current robot state. To enable the relatively weak LLM backbones of open-source VLAs to perform such reasoning effectively, we design a scalable pipeline for synthetically generating embodied CoT training data for large robot datasets. Concretely, we use powerful pre-trained foundation models to generate the reasoning supervision for our policies.

Our experiments show that by training state-of-the-art VLAs to reason before predicting actions, we can substantially boost their ability to perform challenging generalization tasks. Our approach increases the absolute success rate of OpenVLA [7], the current best-performing open-source VLA, by 28% across a suite of robot manipulation tasks that involve generalization to new objects, scenes, viewpoints, and instructions without any extra robot training data. Beyond raw performance improvements, we find that embodied CoT makes policy failures more interpretable and allows humans to easily correct its behavior by modifying faulty reasoning chains via natural language feedback.

## 2  Related Work

**Scaling robot learning.** A long-standing goal of robot learning is to train policies that can generalize to a wide range of unstructured real-world environments. Towards this goal, recent works have explored training "generalist robot policies" [10, 11, 12, 13, 14, 15, 16, 17] on diverse robot datasets [18, 19, 10, 20, 21, 22, 23, 24, 13, 14, 25, 15, 26, 6]. As a result of their diverse robot

training datasets, many of these policies can be prompted in natural language to solve various manipulation tasks, and some generalist policies can even control multiple robot embodiments [16, 27, 6]. Importantly, these works demonstrate that training robot policies on large and diverse datasets is a promising approach towards improving policy robustness and generalization ability.

**Vision-language models for robot generalization.** In a push towards generalization far beyond what is observed in robot datasets, the recent development of strong, open-source vision-language models that learn visuo-linguistic representations [28, 29], generate images from text [30], or generate text in response to images and prompts [31, 32, 33, 34, 35] have resulted in a large number of works that explore the integration of such models into robot learning pipelines, e.g., to generate goals [36], to provide reward signals [37, 38, 39], or to learn visual state representations [40, 41, 42]. Since collection of the aforementioned large-scale robot datasets is challenging, using models pre-trained on Internet-scale data is an appealing alternate path towards robust robot policies that can act in a variety of unstructured real-world environments. Most relevant to our work are recent approaches for integrating pre-trained vision-language models into learned robot policies. While some works use strong structural priors in their policies to enable this integration [43, 44, 45], vision-language-action models (VLAs) have recently been proposed as a simple yet scalable alternative [5, 6, 7], achieving state-of-the-art performance for generalist robot policies [7] and showing impressive levels of generalization to new objects and scenes. However, existing VLAs do not sufficiently leverage some of the most appealing properties of the underlying language and vision-language models, specifically their ability to *reason* through the steps required to solve a given task.

**Reasoning for language and control.** Such step-by-step reasoning is a key ingredient for the ability of large language models (LLMs) to solve a wide range of complex tasks. Prompting LLMs (directly [46] or with in-context examples [8]) to "think step-by-step" about the problem before formulating an answer can significantly improve their performance, with such chain-of-thought reasoning techniques becoming standard practice in language modeling and (vision-)language model training [9, 47]. A number of works have explored similar techniques in the context of high-level task planning for robotics [48, 49, 50, 51, 52, 53, 54, 55, 56]. These approaches use pre-trained or fine-tuned LLMs to decompose tasks into high-level sub-tasks, but rely on pre-trained low-level policies to execute them. However, we argue that (1) careful reasoning can be beneficial for both high-level sub-task reasoning *and* during low-level control and (2) all such levels of reasoning should be strongly grounded in visual observations of the scene and the agent's state. Thus, in contrast to these prior works and language-only CoT, our approach trains a VLA policy to autoregressively generate CoTs (for high- and low-level reasoning) and actions given input instructions and observations, ensuring that both are firmly grounded in the agent's environment. We empirically confirm that such a formulation is critical to effectively leveraging (V)LM reasoning capabilities for control.

## 3 Preliminaries: Vision-Language-Action Models

Our work leverages VLAs as the backbone for our embodied chain-of-thought policies. VLAs use a simple policy learning recipe: starting from a pre-trained vision-language model, they directly finetune the model to autoregressively predict the next robot action $a$ given the current image observation $I$ and task instruction $T$. To enable this, the continuous robot actions are typically converted to discrete action tokens $\mathcal{T}_a$ in

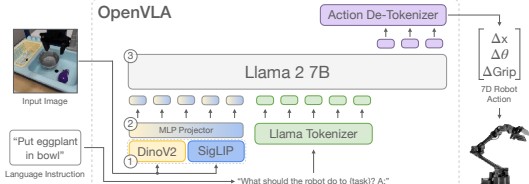

Figure 2: The OpenVLA model. Reproduced with permission from Kim et al. [7].

the vocabulary of the vision-language model via a per-dimension action discretization scheme that assigns each continuous value to one of 256 bins [5, 7].

In this work, we use the recently released OpenVLA model [7] (see Fig. 2), since it achieves state-of-the-art performance and is fully open-source. The model builds on the Prismatic VLM [35] and consists of a fused visual encoder that combines pre-trained SigLIP [57] and/or DinoV2 [58] features and a Llama 2 7B [59] LLM backbone. During training, input images are encoded into visual token embeddings using the pre-trained vision encoders, the task instruction is mapped to task tokens using Llama 2's text tokenizer, and the model is trained to map these inputs to the target action tokens with

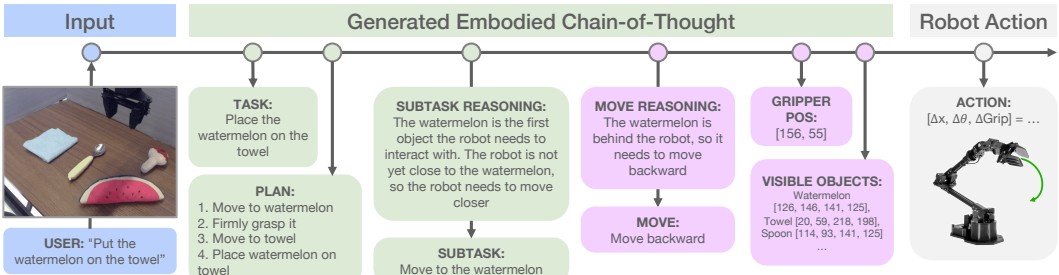

Figure 3: Steps of our embodied chain-of-thought reasoning. We interleave several intermediate reasoning steps into the mapping from inputs to robot actions. **Green**: "standard" linguistic chain-of-thought steps that break a given instruction into the required sub-tasks. **Purple**: *Embodied* chain-of-thought steps that require grounding the policy's reasoning in the scene and robot state. Our experiments show that these grounded reasoning steps are key to improving policy performance with chain-of-thought reasoning.

the standard autoregressive next token prediction objective. We likewise use this objective, but also insert reasoning text tokens in the training sequences such that the policy learns to produce the ECoT reasoning followed by the action in response to the observation and instruction. Next, we will discuss how we can improve upon this conventional VLA training recipe by enabling the VLA to reason through the task at hand before deciding which action to take.

## 4 Embodied Chain-of-Thought Reasoning for Visuomotor Policies

We now discuss our approach for training VLAs to perform embodied chain-of-thought reasoning about plans, sub-tasks, motions, and visual features before predicting the next robot action (see Fig. 1). Unlike many proprietary large language models, the relatively small LLM backbones used in current VLAs struggle to perform involved reasoning when simply prompted to think step-by-step [8]. Instead, we propose to explicitly train VLA models to perform embodied CoT reasoning. Concretely, we label data from existing robot datasets post-hoc with reasoning chains filled with features extracted from various pre-trained models and use the resulting dataset of observation-reasoning-action tuples for training. In practice, we ensure that all elements of the generated reasoning data can be represented as strings, such that we can use the Llama 2 text tokenizer to translate them into reasoning tokens. Then, we train the VLA to autoregressively predict these tokens, directly followed by action tokens.

While this approach is conceptually simple, its implementation requires answering multiple key questions: (1) Which reasoning steps are suitable for guiding policies in solving embodied robot manipulation tasks (Fig. 3)? (2) How can we generate training data for these reasoning steps at scale on existing robot datasets (Section 4.2)? Another practical consideration arises *after* training, while using ECoT policies for robot control: carefully reasoning through each action can significantly slow down policy inference. We discuss solutions to these problems in the following sections.

### 4.1 Designing Embodied Chain-of-Thought Reasoning Steps

Our goals when designing the steps of our embodied chain-of-thought reasoning chains are twofold: encourage the model to (A) reason through the required high-level steps of the task at hand and determine which step needs to be executed next, and (B) increasingly ground this reasoning in lower-level features of the scene and robot state before predicting the robot action.

We visualize the ECoT reasoning steps that we train the VLA to perform for an example task in Fig. 3. From left to right, the model is trained to first rephrase the task instruction (**TASK**) and predict a high-level plan of steps for achieving the instructed task (**PLAN**). Next, it reasons through which of the sub-tasks should be executed at the present step (**SUBTASK**), a task which requires understanding the current state of the scene and robot. Then, the model predicts an even lower-level language command like "move left" or "move up" (**MOVE**) that is closely related to the low-level actions the robot needs to execute. Finally, we ask the model to predict precise, spatially grounded features that describe the scene and thus force the model to pay close attention to all elements of the input image– specifically, the pixel position of the robot end effector (**GRIPPER**) and the names and bounding box pixel coordinates of all objects in the scene (**OBJECTS**).

While we believe that our choice of reasoning tasks and their order is well-aligned with our intuition for a sensible step-by-step solution to the task, we by no means exhaustively explored all possible

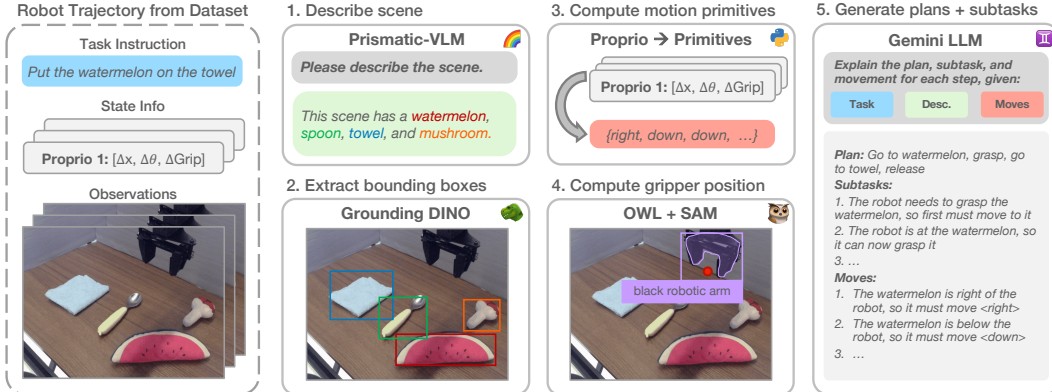

Figure 4: Our pipeline for generating ECoT data at scale for a given robot dataset. We use a Prismatic VLM [35] to create a scene description (**1**), and Grounding Dino [29] to detect bounding boxes for all objects (**2**). We then compute templated motion primitives from low-level robot states (**3**) and the robot gripper position using OWLv2 [60] and SAM [61] (**4**). Finally, we use Gemini [62] to create synthetic reasoning chains (**5**).

reasoning tasks. Testing alternative tasks and task orderings, and finding ways to automatically determine sensible reasoning chains are important directions for future work.

## 4.2 Generating Embodied Chain-of-Thought Data at Scale

While human annotations are the gold standard, it is impractical to get them for large robot learning datasets [6], which consist of millions of transitions. Thus, we instead propose to leverage pre-trained vision and/or language foundation models to automatically generate ECoT training data, akin to synthetic data generation in NLP [63]. We give an overview of our data generation pipeline in Fig. 4.

For a given image-instruction pair, we first prompt a Prismatic-7B VLM [35] to describe the scene. We then concatenate the instruction and this description and input them into grounding DINO [29], an open-vocabulary object detector, to detect all relevant objects and their bounding boxes with the corresponding language snippet labels from the input text. See App. A for examples and details.

Next, we generate the per-step motion primitives in **MOVE** (e.g., "move left"). Following Belkhale et al. [64], we use robot proprioception to determine the movement for the next 4 time steps (assuming a fixed camera), and translate this into one of 729 templated movement primitives (see App. B). We use OWLv2 [60] and SAM [61] to detect 2D end effector positions in the training images (**GRIPPER**) paired with 3D positions extracted from the robot state to fit a robust estimate of the projection matrix using RANSAC [65]. We then use the 2D projections of the robot end-effector position for our training. This is repeated for each trajectory, eliminating the need to assume fixed camera parameters.

To generate the final reasoning chain, we feed each episode's task instruction, scene description, and per-step movement primitives into Gemini 1.0 [62] and prompt it to produce both a high-level plan of sub-tasks in accordance with the task instruction and observed movement primitives and the current sub-task for each step. We also ask it to briefly explain the primitive movement and chosen sub-task in each step, which we include in the ECoT training data. We run our data generation pipeline on the complete Bridge v2 dataset [13], with more than 2.5M transitions, over the course of 7 days.

## 4.3 Efficient Chain-of-Thought Inference for Robot Policies

Inference speed is a key challenge for ECoT policies. They needs to predict around 350 reasoning tokens per timestep, compared to OpenVLA's 7 action tokens. We explore a simple solution for speeding up inference: as our policy learns next token prediction, it can continue arbitrary input reasoning prefixes; we thus can keep parts of the reasoning chain like the high-level plan or the current sub-task fixed for multiple steps. *Encoding* previously predicted tokens is much faster for Transformer-based policies than *generating* them. We compare two such strategies: (1) *synchronous* execution, where we predict the high-level reasoning every $N$ steps, and (2) *asynchronous* execution, in which one ECoT policy instance continually updates the high-level reasoning chains, while a second policy instance uses the most recent reasoning chain to predict low-level reasoning steps and actions. We report the performance and inference speed for all approaches in Section 5.5. Note that these speed-ups are orthogonal to widely used approaches for improving throughput of LLMs. We explore one such approach – compiling with TensorRT-LLM [66] – in App. F.

Table 1: **Comparison of success rates for OpenVLA, RT-2-X, and ECoT** across two scenes (one with in-distribution camera view and one with out-of-distribution). Mean ± one StdErr. On aggregate, our ECoT policy achieves the highest success rate, improving absolute success rate by 45%, 22%, 19%, and 18% over Octo, OpenVLA, RT-2-X, and naïve CoT respectively in the in-distribution view setting and 48%, 34%, 16%, and 16% in the out-of-distribution view setting.

| Type | Task | Algorithm (ID View) | | | | | Algorithm (OOD View) | | | | |
|---|---|---|---|---|---|---|---|---|---|---|---|
| | | Octo | OpenVLA (Bridge) | RT-2-X | Naïve CoT | ECoT (Ours) | Octo | OpenVLA (Bridge) | RT-2-X | Naïve CoT | ECoT (Ours) |
| **ID** | Put mushroom in pot | 29% | 88% | 94% | 71% | **100%** | 35% | 59% | **76%** | **76%** | 65% |
| | Put spoon on towel | 60% | **90%** | 80% | 60% | 80% | 20% | **80%** | **80%** | 60% | **80%** |
| | Put carrot on plate | 70% | 80% | 90% | 90% | **100%** | 40% | 90% | 90% | **100%** | 90% |
| | Wipe [plate / pan] with towel | 13% | **50%** | 38% | 38% | **50%** | 0% | 50% | 0% | 13% | **63%** |
| **Spatial Relations** | Put mushroom in [left / right / middle] container | 0% | 22% | 17% | 22% | **33%** | 0% | 17% | 22% | 55% | **67%** |
| | Put purple object in [left / right / middle] container | 0% | 28% | 17% | 50% | **56%** | 0% | 22% | 11% | **55%** | 39% |
| | Put [right / left] object on middle object | 0% | 13% | 0% | 50% | **63%** | 0% | 25% | 25% | 50% | **63%** |
| **OOD Objects** | Pick up [screwdriver / hammer / measuring tape / detergent / watermelon] | 30% | 20% | **80%** | 50% | 50% | 30% | 20% | **80%** | 50% | 50% |
| | Move mushroom to [measuring tape / detergent] | 0% | 10% | 70% | 20% | **100%** | 10% | 0% | **90%** | 40% | **90%** |
| | Put mushroom in tall cup | 0% | **80%** | 0% | 70% | 30% | 10% | 20% | 0% | 20% | **30%** |
| | Place watermelon on towel | 20% | 30% | 60% | 60% | **70%** | 50% | 10% | **90%** | 30% | 40% |
| **OOD Instructions** | Pick up any object that is not [yellow / a duck / a sponge / a towel] | 50% | 33% | **58%** | 50% | 42% | 17% | 17% | **67%** | 25% | **67%** |
| | Put the edible object in the bowl | 13% | 25% | 13% | 25% | **88%** | 0% | 13% | 25% | 25% | **100%** |
| | Put the object used for [eating / drinking] on towel | 25% | 38% | 38% | 38% | **75%** | 13% | 0% | 25% | 38% | **75%** |
| | **Aggregate** | 21% ± 3.3% | 44% ± 3.9% | 47 ± 4.0% | 48 ± 4.0% | **66% ± 3.8%** | 16% ± 2.9% | 30% ± 3.6% | 48 ± 4.0% | 48% ± 4.0% | **64 ± 3.9%** |

# 5 Experiments

In this section, we investigate the effectiveness of ECoT for robot control across a range of challenging manipulation tasks. We answer the following questions: (1) Does embodied chain-of-thought reasoning improve the performance of VLA policies (Section 5.2)? (2) Does embodied chain-of-thought reasoning make it easier to interpret and correct policy failures (Sections 5.3 and 5.4)? (3) How can we optimize the runtime efficiency of policies with embodied CoT reasoning (Section 5.5)?

## 5.1 Experimental Setup

**Robot setup and training data.** We evaluate on a 6-DoF WidowX robot arm, a commonly-used setup for evaluating learned robot policies [16, 7]. Given a single 3$^{rd}$ person camera and natural language instruction, the policy predicts end-effector velocity actions to control the robot. Bridge v2 [13] provides a diverse dataset of 60k teleoperated demonstrations. We apply our pipeline for synthetic generation of chain-of-thought data (Section 4.2) on this dataset to obtain our training dataset.

**Evaluation tasks.** We design a suite of challenging evaluation tasks that focus on testing the policies' generalization ability along multiple axes: processing spatial relations, interacting with unseen objects, and performing unseen instructions. All policies are evaluated on the same real-world setups to control for camera angle, lighting, and background. We perform 314 total trials per approach.

**Comparisons.** We compare our policy (**ECoT**) to state-of-the-art VLA policies, namely **Open-VLA** [7], the same model our approach is built upon, but trained *without* chain-of-thought reasoning, and **RT-2-X** [6], a 55B parameter closed VLA policy. To ensure fair comparison, we train the OpenVLA policy on the same dataset we use for training our approach (the Bridge V2 data [13]), denoted as **OpenVLA (Bridge)**. For RT-2-X we cannot control the data distribution in the same way since the model is closed, but it was trained on Bridge V2 data *and additional datasets from the Open X-Embodiment dataset* [6]. Thus, it was trained on *more* data than our approach. We also compare against **Octo** [16], which is also trained on that dataset, but was not fine-tuned from a VLM (i.e., it is not a VLA). Finally, we compare to **Naïve CoT**, a version of our model that only uses *non-embodied* CoT reasoning about sub-tasks akin to conventional CoT reasoning in language modeling (see Fig. 3). This comparison will test the importance of using *embodied* reasoning for VLA policies.

## 5.2 Embodied Chain-of-Thought Reasoning Improves Policy Generalization

We report performance of all approaches on our evaluation set in Table 1. We see that while OpenVLA (Bridge) achieves high performance on in-distribution tasks, it struggles on the hard generalization cases we test. RT-2-X performs better than vanilla OpenVLA (Bridge), potentially due to the larger robot pre-training dataset (note again that OpenVLA and our approach are *only* trained on the Bridge dataset) and the fact that it *co-trains* the policy with Internet-scale vision-language data *and* robot data, while all other approaches only use robot data during fine-tuning.

Importantly, we find that our ECoT policy substantially outperforms OpenVLA (Bridge) across all generalization evaluations but one. This is notable, since both policies have the exact same VLM base

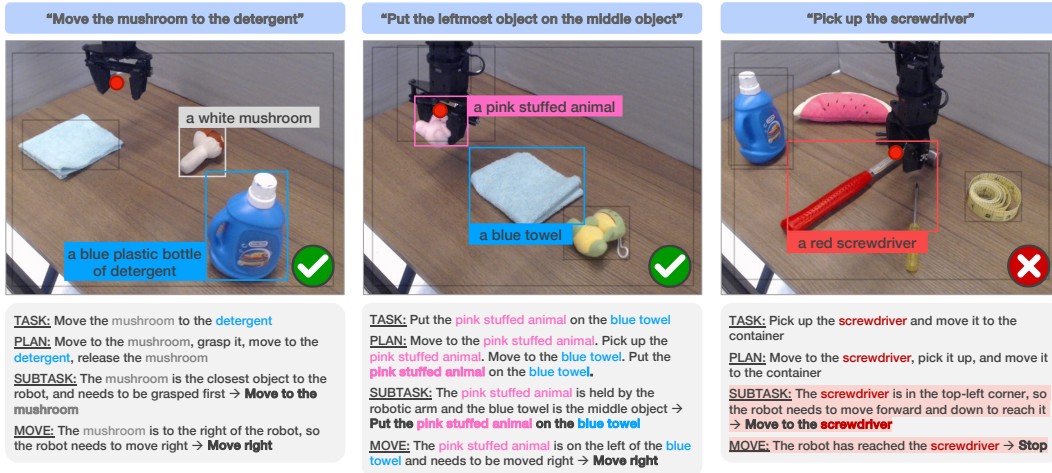

Figure 5: **Qualitative ECoT predictions from our model for two successful trajectories (left, middle) and one failure (right).** Irrelevant bounding boxes are greyed out for readability. Left: high-level reasoning and low-level object segmentations are correct, leading to a successful rollout. Middle: the command is correctly rephrased to refer to specific objects (i.e., "the leftmost object" is identified as the pink toy). Right: the hammer is incorrectly identified as a screwdriver, causing the robot to take inappropriate actions.

model and use the same robot data for fine-tuning. The only difference is in the use of CoT reasoning by our approach. Curiously, our ECoT model even surpasses the performance of RT-2-X in the tested tasks, even though RT-2-X is trained on 10 additional robot datasets and uses a neural network that is 7x larger (55B vs. 7B). Finally, the results in Table 1 show that including *embodied* reasoning about visual inputs and the low-level robot state significantly boosts performance over the "Naïve CoT" ablation of our approach, which only reasons about high-level linguistic features like sub-task plans.

We visualize qualitative examples of our model's reasoning in Fig. 5. The left two examples show that the model successfully breaks down the task into a sequence of sub-tasks and then crucially *grounds* those sub-tasks in the scene by predicting the relevant bounding boxes and gripper position of the robot, before deciding on the next move and concrete low-level robot action. We visualize more chain-of-thought examples from our evaluation tasks in Fig. 8.

### 5.3 Diagnosing Policy Failures Through Inspecting Reasoning Chains

In addition to improving performance, such embodied chain-of-thought reasoning provides a tool for operators to better understand the decisions the policy takes. By inspecting and visualizing the model's reasoning steps, we can discover potential mistakes in the reasoning chain that led to policy failure downstream. For instance, in Fig. 5 (right), the ECoT policy failed to solve the task *pick up the screwdriver*. Inspecting the reasoning chain, we can see that the hammer is incorrectly identified as a screwdriver, causing the robot to reach for that instead. Note that inspecting reasoning chains is not "bullet-proof" for interpreting end-to-end policy failures: the model *could* predict a particular plan and then still deviate from it when choosing the final action. However, in practice we find that reasoning chains often correlate strongly with the executed actions. We provide more examples for diagnosing policy failures via its reasoning chains in Fig. 8.

### 5.4 Chain-of-Thought Reasoning Enables Interactive Policy Correction

Training a policy to reason through a task step-by-step provides a powerful mechanism for humans to interact with the policy and *correct* its behavior: instead of needing teleoperation equipment to provide direct robot action feedback like in DAgger [67], users can simply modify its reasoning chains via natural language feedback to correct the policy's behavior. Prior work used carefully designed architectures explicitly trained to support such correction via language [68, 69], but we test whether similar capabilities emerge naturally by training VLA policies to perform ECoT reasoning.

We rerun evaluations of our ECoT policy on some of the most challenging tasks from Table 1 (put mushroom in cup, pick up out-of-distribution object, and pick up non-yellow object), in which our policy *without human intervention* achieved an average success rate of only 32%. We allow a human operator to interrupt policy execution *once* over the course of the episode and provide natural

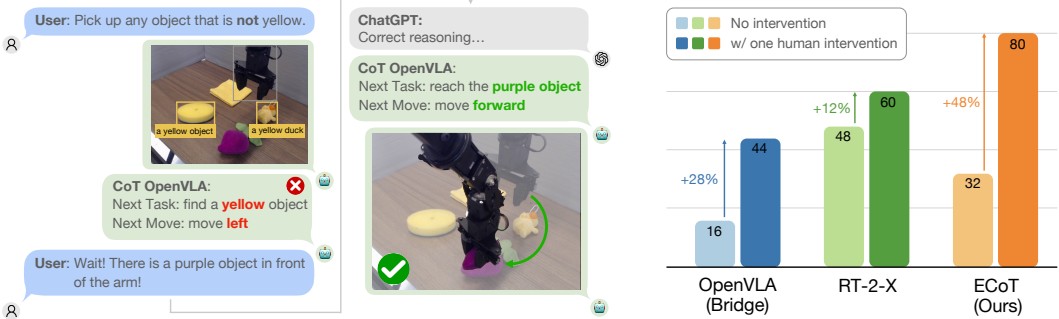

Figure 6: Embodied chain-of-thought training enables interactive human policy correction in natural language. **Left**: given a human intervention in natural language, we use ChatGPT to correct our model's reasoning chains. **Right**: our embodied chain-of-thought policy can benefit from a human language intervention most, increasing success rate by 48% on our most challenging evaluation tasks.

language feedback (e.g. "no, the screwdriver is in the back right corner", "release the mushroom now!", or "the cup is tall") (see Fig. 6). Then, we use ChatGPT to adapt our model's reasoning chain based on the language feedback, prompting it to produce a corrected reasoning chain (see Fig. 11 for the exact prompt used). Finally, we feed this corrected chain back into our policy and continue execution, holding the corrected reasoning chain fixed for 5 steps.

The results in Fig. 6 (right) show that our ECoT policy can make effective use of the human language intervention, increasing its success rate by 48%. In contrast, we evaluate the vanilla OpenVLA policy and RT-2-X in the same way, providing each with a single human language correction per rollout, but find that neither of them can benefit from the human intervention to the same degree (for both we also use ChatGPT to incorporate the intervention into the original task instruction to allow for fair comparison). This is because standard VLAs that directly map from images and instructions to actions do not expose any part of their internal decision-making process. Not only does this make "debugging" incorrect behaviors difficult, it also only permits changing the prompt for interventions.

### 5.5 Efficient Chain-of-Thought Inference

We compare approaches for accelerating ECoT policy inference (see Section 4.3) to naïvely running the full generation every step of execution in Table 2. We also report the speed-up that is achieved by both proposed approaches from Section 4.3 vs. naïve execution. Both approaches achieve inference speed improvements while at least matching performance, with asynchronous execution achieving the largest speed-up at the cost of doubling the compute required at inference time (since two policy instances are running in parallel). We use the 5-step freeze approach for the main results presented in Table 1, since it provides the best performance-speed tradeoff. We used a small task subset (put mushroom in pot, move mushroom to detergent/measuring tape, and put the left/right object on the middle).

Table 2: Performance of accelerated CoT inference approaches and their relative speed-ups over the naïve approach, averaged across 3 tasks (25 trials total).

| | Success | Speed-Up |
|---|---|---|
| Naïve | 63% | – |
| 5-Step | 72% | +24% |
| Async | 65% | +40% |

## 6 Discussion and Limitations

In this work, we demonstrate that training VLA policies to perform chain-of-thought reasoning can substantially increase their performance without the need to collect additional robot training data. Instead of simply applying the CoT recipe from language modeling, our experiments underline the importance of adding reasoning steps that are strongly grounded in the task, scene, and robot state. While our results are encouraging, our approach has several limitations. First, our model does not adapt the *structure* of its reasoning chains to the task at hand; it always performs *all* steps of reasoning in the fixed order we chose. A more effective strategy may involve executing only a subset of reasoning steps based on the robot and scene state, and future work can explore directly optimizing the model to pick the best reasoning steps. Additionally, scaling the ECoT training to a larger subset of the OXE dataset [6] will improve transfer of ECoT capabilities to more robots.

**Acknowledgments**

We would like to thank Sidd Karamcheti for his help with using the Prismatic VLM and OpenVLA codebases. We also thank Google DeepMind for providing Alpha API access to the RT-2-X model. This research was partly supported by ONR under N00014-20-1-2383, NSF IIS-2150826 and IIS-2246811 and the AI Institute. Michał Zawalski was funded by the Polish National Agency for Academic Exchange (PPN/STA/2021/1/00079/U/00001).

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

# A Grounding DINO Detections and Prismatic Descriptions

We provide example scene descriptions provided by Prismatic VLM and bounding boxes provided by Grounding DINO in Fig. 7. We filter the predictions based on the provided confidence score, only keeping detections with a box- and text-confidence larger than 0.3 and 0.2 respectively to use for the **OBJECT** features.

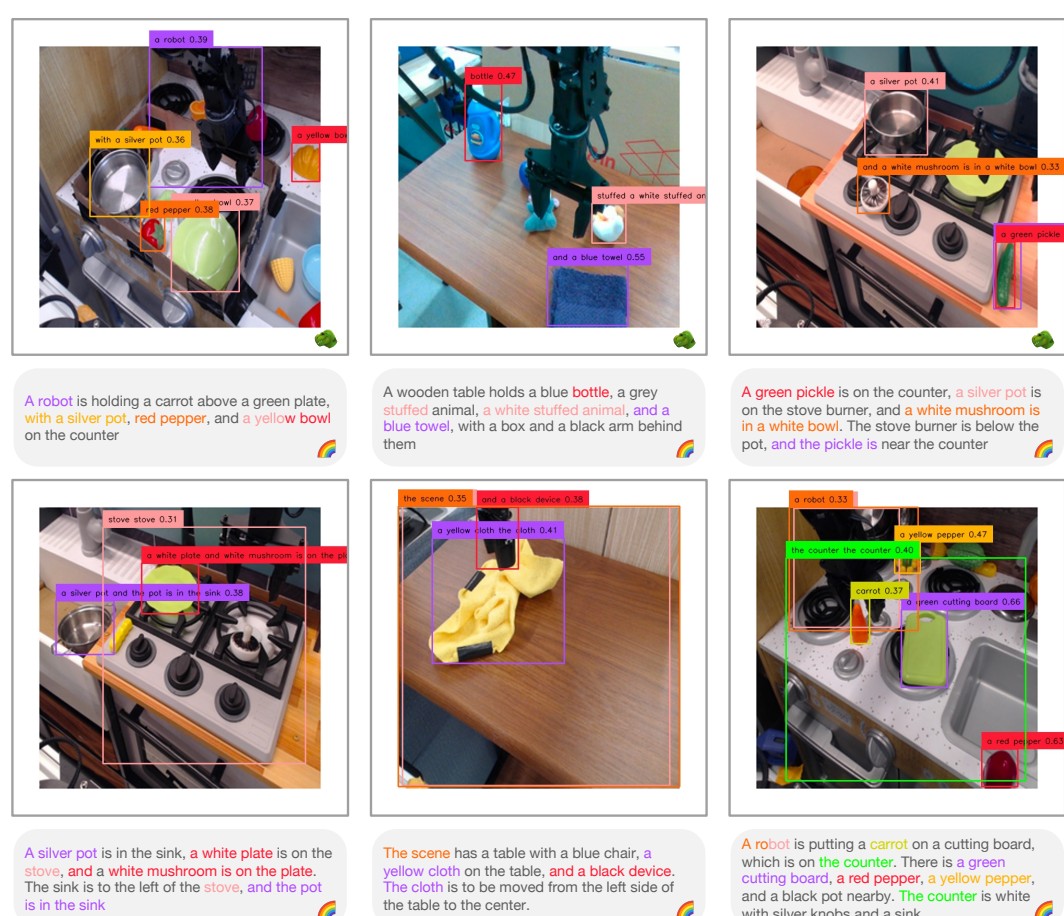

Figure 7: Examples of captions of observations from the Bridge dataset as generated by our Prismatic VLM, as well as associated bounding boxes generated by Grounding DINO.

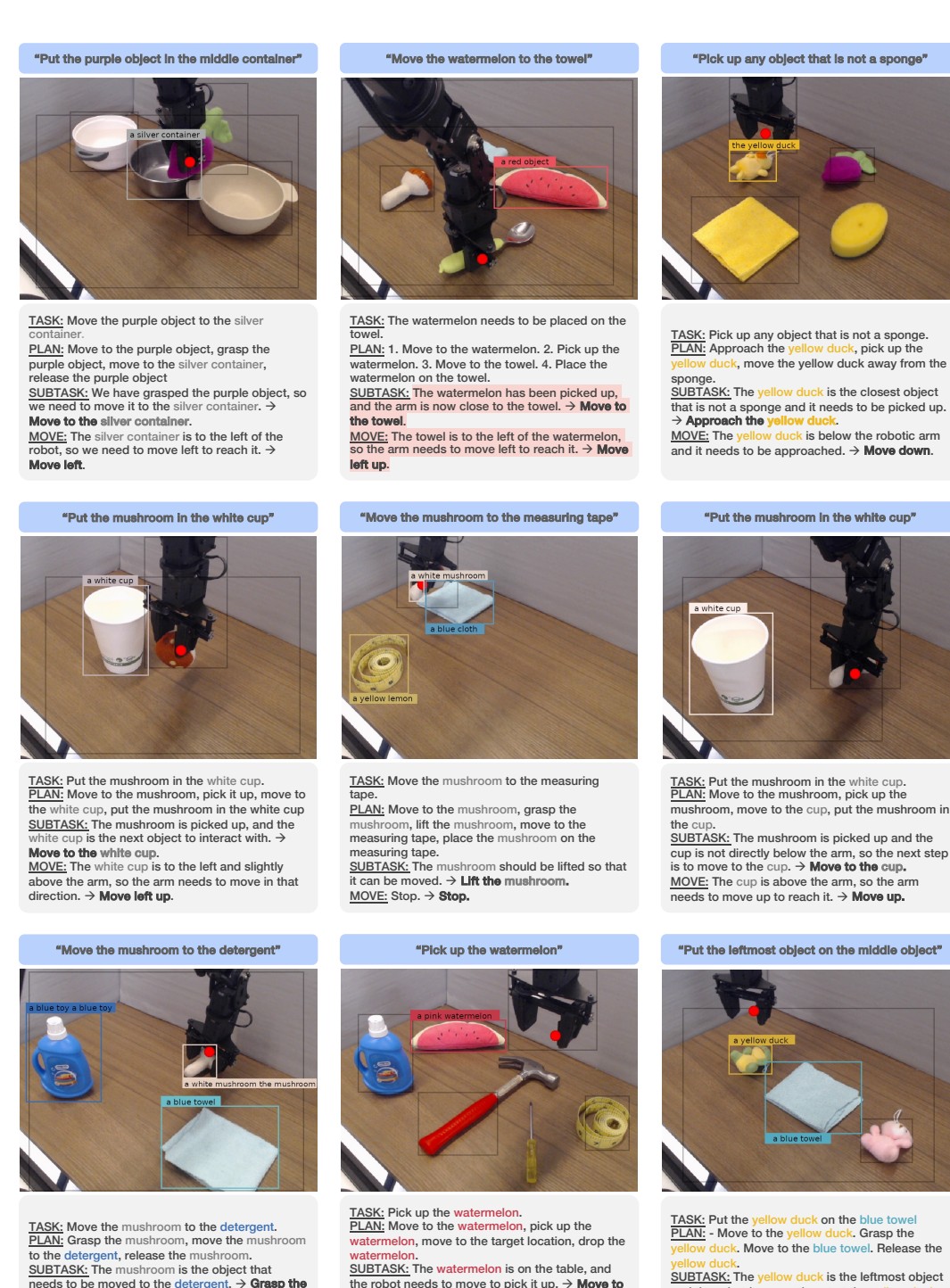

Figure 8: More qualitative examples of successful and failed chain-of-thought reasonings.

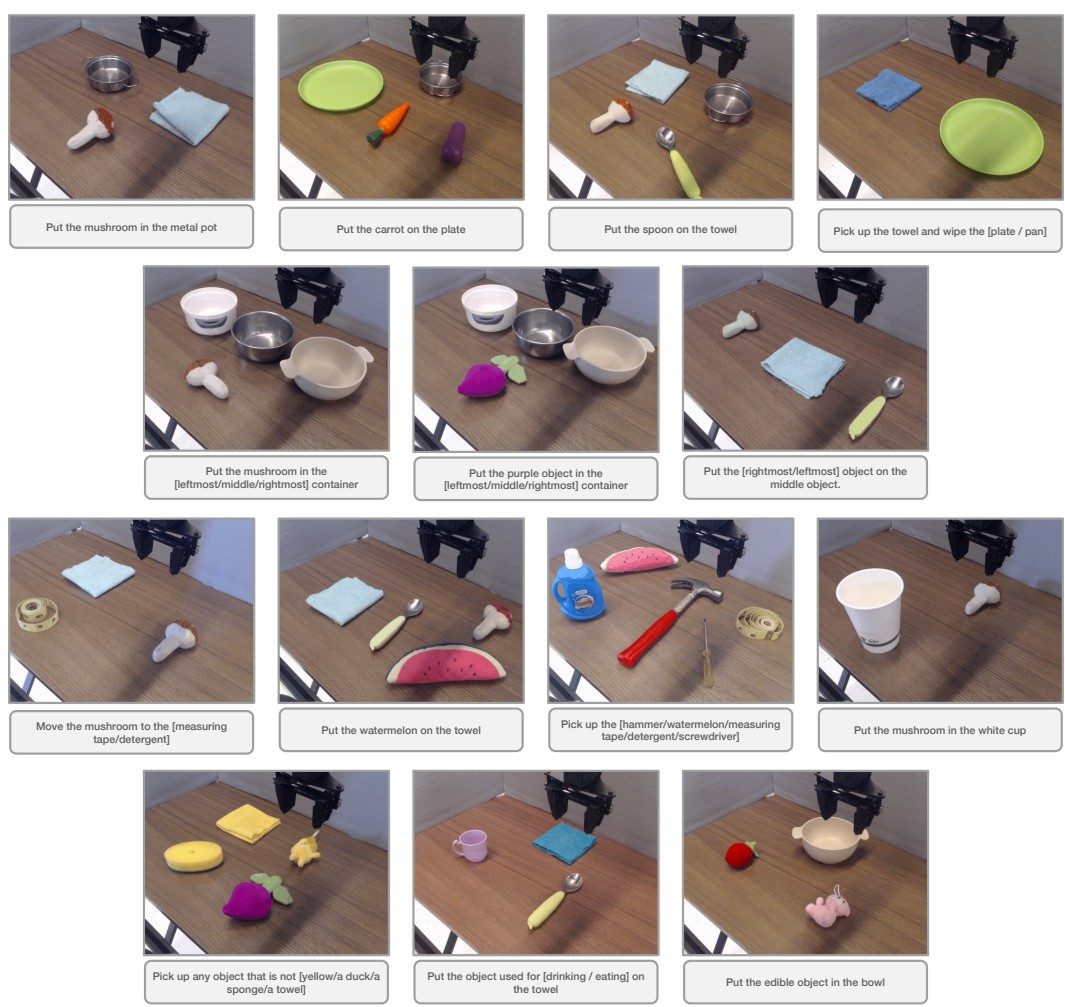

Figure 9: Example starting scenes and associated prompt for all task types.

# B  List of Movement Primitives

To classify a movement, we take the difference between the current state of the robot and its position four steps ahead. Based on the axes where the difference exceeds a threshold of 0.03, we assign it a label of the following form:

```
move [forward/backward] [left/right] [up/down], tilt [up/down], rotate
        [clockwise/counterclockwise], [close/open] gripper
```

Whenever the movement in a certain axis is below the threshold, we omit its block for simplicity. For instance, if the robot is just moving left, the label is move left. If no movement is detected, the label is stop.

While technically it results in $3^6 = 729$ possible labels, only 54 are used in more than $0.1\%$ of cases:

1. stop (26.9%)
2. close gripper (10.8%)
3. open gripper (7.2%)
4. move down (6.8%)
5. move left (6.6%)
6. move right (6.1%)
7. move up (5.7%)
8. move forward (3.0%)
9. move backward (2.4%)
10. move up, open gripper (2.1%)
11. move forward right (1.1%)
12. move up, close gripper (1.0%)
13. move backward left (1.0%)
14. move forward left (0.9%)
15. move left down (0.8%)
16. move down, close gripper (0.8%)
17. move right down (0.8%)
18. move left up (0.8%)
19. move right up (0.8%)
20. move right, rotate clockwise (0.8%)
21. move left, rotate counterclockwise (0.8%)
22. move backward right (0.8%)
23. rotate counterclockwise (0.7%)
24. move down, open gripper (0.7%)
25. rotate clockwise (0.7%)
26. move forward down (0.7%)
27. move up, rotate clockwise (0.5%)
28. move up, rotate counterclockwise (0.5%)
29. move backward up (0.5%)
30. move left, rotate clockwise (0.3%)
31. move backward down (0.3%)
32. move right, open gripper (0.3%)
33. move forward up (0.3%)
34. move left, open gripper (0.3%)
35. move right, rotate counterclockwise (0.3%)
36. move backward, open gripper (0.2%)
37. move down, rotate clockwise (0.2%)
38. move down, rotate counterclockwise (0.2%)
39. move forward, rotate counterclockwise (0.2%)
40. move forward, rotate clockwise (0.2%)
41. move forward, open gripper (0.2%)
42. move right, close gripper (0.2%)
43. move backward, rotate clockwise (0.2%)
44. move backward, rotate counterclockwise (0.2%)
45. move left, close gripper (0.2%)
46. move backward right, rotate clockwise (0.1%)
47. move backward left, rotate counterclockwise (0.1%)
48. move right up, open gripper (0.1%)
49. move right up, close gripper (0.1%)
50. move backward, close gripper (0.1%)
51. rotate clockwise, close gripper (0.1%)
52. rotate counterclockwise, close gripper (0.1%)
53. move left up, open gripper (0.1%)
54. move forward right, rotate clockwise (0.1%)

## C   Prompts

We now provide all the prompts used for data generation and policy language conditioning.

For using generating scene descriptions with Prismatic (step 1 in Fig. 4), we use the prompt: "Briefly describe the things in this scene and their spatial relations to each other." We prepend "The robot task is: [*TASK*]." if the given demonstration trajectory contains a corresponding task instruction (where we ensure that said instruction contains at least one space character to remove noisy instructions).

We provide the prompt for Gemini data labeling (step 5 in Fig. 4) in Fig. 10. We re-run generation if Gemini fails to produce reasonings of the correct format.

The prompts used for our language-conditioned policies are provided in Fig. 9, along with example starting scenes for the associated tasks. For the OpenVLA-based policies, said prompts are inserted into the template provided by the original authors [7]: "A chat between a curious user and an artificial intelligence assistant. The assistant gives helpful, detailed, and polite answers to the user's questions. USER: What action should the robot take to [*PROMPT*]? ASSISTANT:". The agent then generates reasoning text (if trained to do so) and an action.

We provide the prompt used for human interventions with ChatGPT in Fig. 11.

## D   Clutter Evaluations

We provide additional evaluations of how our policy compares against OpenVLA in settings with significant clutter (9 or more objects per scene). We evaluate two rollouts for eight tasks in the four categories used in Table 1: in-distribution (put mushroom on cloth, put corn in pot), spatial relations (put mushroom in [leftmost / rightmost] container, put mushroom in [leftmost / rightmost container]), out-of-distribution objects (move mushroom to measuring tape, place watermelon on pan), and out-of-distribution instructions (put edible object on towel, put animal in pot). Note that some tasks are not from the original tasks in Table 1, which we chose to make them more amenable to evaluation in cluttered scenes. We show example starting arrangements for each task in Fig. 12.

We find that our ECoT policy achieves 75% success rate while OpenVLA only achieves 25%, indicating that reasoning may aid in robustifying against clutter. Qualitatively, we note that many failure cases are exacerbated for OpenVLA in the presence of clutter, while our ECoT policy still enjoys the same benefits noted in the default task suite. Namely, OpenVLA often gets distracted by commonly-represented clutter objects (like the mushroom) while failing to reason about object semantics, which our approach deals with well (e.g., successfully identifying toy foods and stuffed animals when queried to pick the edible objects and animals respectively, even though such wordings are rare or non-existent in our training data).

## E   Task Objects

We provide visualizations of example representative starting setups for all tasks in Fig. 9. The tasks involve the following objects:

1. *Put mushroom in metal pot*: mushroom, shallow metal pot, light blue towel, spoon

2. *Put carrot on plate*: carrot, eggplant, shallow metal pot, green plate

3. *Put spoon on towel*: spoon, light blue towel, mushroom, shallow metal pot

4. *Wipe [plate / pan]*: plate, pan, dark blue towel, yellow towel

5. *Put the [purple object / mushroom] in the [leftmost / middle / rightmost] container*: purple beet, mushroom, tan plastic bowl, metal bowl, white plastic bowl

6. *Put [rightmost / leftmost] on middle*: mushroom, spoon, pink bear, green frog, light blue towel

7. *Move mushroom to [measuring tape / detergent]*: measuring tape, detergent, mushroom, light blue towel

8. *Put the watermelon on the towel*: watermelon, mushroom, spoon, light blue towel

9. *Pick up the [hammer / watermelon / measuring tape / detergent / screwdriver]*: hammer, watermelon, measuring tape, detergent, screwdriver

10. *Put mushroom in white cup*: tall white cup, mushroom

11. *Pick up any object that is not [yellow / a duck / a sponge / a towel]*: yellow duck, yellow towel, sponge, purple beet

12. *Put the object for [drinking / eating] on the towel*: dark blue towel, spoon, cup, mushroom

13. *Put the edible object in the bowl*: tomato, pink bear, carrot, yellow duck, banana, green frog, corn, monkey

We frequently use the mushroom and light blue towel as distractors, as they are well-represented in our training data, and so non-reasoning policies often erroneously pick them up or place things on them when instructed otherwise, especially when dealing with the less common or out-of-distribution objects (measuring tape, detergent, watermelon, hammer, screwdriver, white cup).

## F    Additional Analysis and Experiments

**Can we improve inference speeds by compiling the language model backbone?** As our model shares an architecture with OpenVLA [7], which likewise is fine-tuned from a Prismatic VLM [35], the policy itself has a vision and language model backbone. The latter is where much of the inference time is spent: it is a fine-tuned Llama 2 [59] model that has to generate $\sim 300$ chain-of-thought reasoning language and action tokens in response to 256 soft prompt image embedding tokens. However, since it does use this popular language model architecture, our policy can benefit from the numerous recent developments in improving LLM inference speeds.

We show this by compiling our ECoT VLA's language model backbone with TensorRT-LLM [66]. Using FP8 quantization [70], we were able to lower the inference time of generating a full reasoning chain followed by actions from $\sim 5$ seconds to $\sim 1$ second (or $\sim 0.8$ seconds with freezing, from $\sim 2$ seconds). We note that further optimizations (both with TensorRT-LLM or with other techniques, like speculative decoding [71]) are possible and that the policy can be made smoother with techniques like action chunking [2], but we leave these approaches to future works.

**Can we improve speed and interpretability of the ECoT reasoning?** We test two modifications to the structure of our reasoning chains. First, we move the bounding box generations earlier in the chain, right after the plan. This way, we can keep the bounding boxes fixed in our N-step inference (see Section 4.3. Since the bounding box generation represents a significant fraction of the predicted tokens, this change can speed up ECoT inference by $30 - 50\%$ in our experiments. Secondly, we train the model to autoregressively predict the next four future gripper positions, in addition to the current one. Not only does this gives operators a rough visualization of what the ECoT policy expects its motion to be in the future, but it serves as an (albeit imperfect) proxy indicating how our policy would behave. This will be important for the following experiments on other robot embodiments, for which we do not have the ability to run real world rollouts.

We evaluate this policy (and all subsequent ones) on a large subset of tasks on the out-of-distribution view station, totaling 106 trials per policy. We find that, while this frozen bounding box policy does perform worse than our base ECoT model, it nonetheless outperforms all baselines (Octo, OpenVLA (Bridge), and RT-2-X), as shown in Table 3. Thus, due to its relative higher speed and ability to visualize rollouts, we adopt this structure for all subsequent experiments.

**Does co-training with vision-language data help?** During VLA fine-tuning, both ECoT and OpenVLA lose the base VLM's ability to respond conversationally to natural language questions. This can be remedied by *co-training* the VLAs with vision-language training data in addition to robot action data. Prior work found that such co-training can improve VLA capabilities [5]. We test performance of an ECoT model co-trained with robot data *and* the vision-language training dataset of the base Prismatic VLM [35] at a 3:1 ratio. Qualitatively, we find that the co-trained model indeed retains its ability to answer questions in chat format in addition to robot control. We compare the performance of this model with our base ECoT model across a large subset of the robot control tasks in Table 1 (in-distribution view), totaling 106 trials per model. The results in Table 3 suggest that co-training does not lead to measurable performance improvements on our evaluation tasks. Anecdotally, we find that the co-trained model can more reliably recognize celebrities and improve performance on tasks like "bring coke can to Taylor Swift" (4/4 successes vs. 0/4 for our base ECoT).

Table 3: **Success rate of ECoT trained with various design choices**, as evaluated on a large subset of trials on the harder out-of-distribution view setting. While our base policy performs the best on aggregate (69%), the other approaches achieve higher performance on certain tasks. All policies in this table outperform OpenVLA (Bridge), RT-2-X, and Octo's performances on the same trial subset (29%, 46%, and 14% aggregate success rates respectively).

| Task | Base ECoT | Frozen Bbox ECoT | Co-trained ECoT | Fine-tuned ECoT |
|---|---|---|---|---|
| Put mushroom in pot | 57% | 86% | 86% | 86% |
| Put spoon on towel | 80% | 60% | 40% | 80% |
| Put carrot on plate | 83% | 67% | 100% | 100% |
| Wipe [plate / pan] with towel | 75% | 50% | 25% | 25% |
| Put mushroom in [left / right / middle] container | 89% | 55% | 11% | 44% |
| Put purple object in [left / right / middle] container | 44% | 67% | 44% | 67% |
| Put [right / left] object on middle object | 63% | 75% | 75% | 75% |
| Pick up [screwdriver / hammer / measuring tape / detergent / watermelon] | 50% | 60% | 80% | 70% |
| Move mushroom to [measuring tape / detergent] | 90% | 90% | 100% | 60% |
| Put mushroom in tall cup | 20% | 20% | 20% | 40% |
| Place watermelon on towel | 60% | 0% | 20% | 40% |
| Pick up any object that is not [yellow / a duck / a sponge / a towel] | 67% | 58% | 42% | 17% |
| Put the edible object in the bowl | 100% | 75% | 50% | 38% |
| Put the object used for [eating / drinking] on towel | 75% | 38% | 50% | 25% |
| **Aggregate** | 69% | 60% | 56% | 54% |

**Does ECoT capability transfer to other robots?** We test whether fine-tuning a *generalist* VLA policy with ECoT data can transfer ECoT reasoning between robot embodiments. Concretely, we use the official checkpoint of the OpenVLA-7B model [7], which was trained on a mix of 27 robot datasets. We continue training the released checkpoint on this mix, but replace the original BridgeData V2 dataset with our generated ECoT dataset. As a result, approximately 13% of the training data is ECoT data. We make two key findings. First, fine-tuning a pre-trained VLA to perform ECoT reasoning is substantially faster than training an ECoT VLA from the base VLM. We observe that within 20k training steps the fine-tuned model nearly matches the performance of our original ECoT model trained for 80k steps (Table 3). Qualitatively, we even observe comparable performance after only 2500 steps. This represents a 4x and 30x reduction in required compute respectively.

Secondly, we find that the fine-tuned model can perform ECoT reasoning on other robot embodiments than it has been trained for, simply by prompting it with the beginning of a ECoT sequence ("TASK:") (Fig. 13). It recognizes robot grippers, objects and their positions, and predicting future gripper movements, despite the large differences in robot appearance, scene layout and camera setup. This result is surprising, since we only provided ECoT training data for a single robot embodiment: the WidowX robot in the BridgeData V2 dataset. We hypothesize that the VLM pre-training enables the model to generalize the concepts of robot end-effector position and movement, and object idendity and positions between robots and scenes. We also tried rolling out the fine-tuned ECoT model in the SIMPLER real-to-sim environments of [72] on the Google Robot tasks, while prompting for ECoT prediction as described above. However, we found that the ECoT model suffered from the real-to-sim domain gap of the SIMPLER environments, producing more faulty reasoning chains than on real Google robot images, and thus not improving overall performance compared to an OpenVLA baseline without ECoT.

# G    Extended Qualitative Analysis of Reasoning Behaviors

Along with Fig. 5, we provide further qualitative examples of successful and failed ECoT reasonings in Fig. 8. We find that most failures are due to some kind of semantic scene misunderstanding, the most common form being incorrect bounding box labels (as shown in the rightmost example in Fig. 5). However, even when bounding boxes are somewhat correct (see top middle example in Fig. 8 where the watermelon is labeled as a red object and the spoon has no label), the model still erroneously believes the watermelon has been grasped, when it actually grasped the spoon. We also note that our model can often handle noisy labels: the bottom right example in Fig. 8 labels the green and yellow frog as a "yellow duck," however the reasoning referring to it still aligns with the specified task. Finally, we note that failures from incorrect plans or subtasks (but correct scene understanding) are typically uncommon, especially because the plans are re-generated after freezing and are often corrected in the process.

# H    Extended Discussions of Design Choices and Alternatives

We now discuss some the advantages and disadvantages of various alternative design choices and optimizations.

## H.1    Inference Optimizations

Performing ECoT inference is significantly slower than "standard" VLA counterparts. Specifically, OpenVLA and RT-2-X achieve control frequencies of 3 and 2 Hz respectively, while our freezing approach achieves 0.5. However, we note that this slower inference speed is nonetheless empirically sufficient for outperforming these standard VLAs (see Table 1). Additionally, there are standard inference speed optimizations that can still be implemented in conjunction with the ones we tried. Most prominently, as mentioned in Section 4.3, TensorRT-LLM [66] claims to enable inference speeds of over 1000 tokens per second for Llama 2, which acts as the language model backbone of our ECoT policy. Given that our ECoT reasoning chains are typically around 250 tokens, this should enable control frequencies on par with non-reasoning VLAs. However, as our primary contribution is to show how reasoning aids in robot control policies, we leave such optimizations to future works.

## H.2    Alternative Reasoning Steps and Orders

As shown in App. F, our approach can be applied for other reasoning step features or orderings while staying more performant than our baselines (see Table 3). Such alternative reasoning steps can give benefits like better interpretability for easier debugging, as demonstrated with our future gripper position feature in Fig. 13. Some other features include open vocabulary 3D bounding boxes, which could be extracted by systems like FM-OV3D [73] (albeit requiring depth estimates, which our training data is not annotated with, but can be extracted using other foundation models). Likewise, PaliGemma [74] can be used for open-vocabulary semantic segmentation (instead of our current 2D bounding boxes), which also has the benefit of providing such segmentation maps in tokenized textual formats. As more foundation models are released for different semantic tasks, we expect that the type of reasoning steps our model can learn via distillation to only increase.

## H.3    External Modules

We choose to train our model to perform all reasoning steps end-to-end. However, in principle, some of these reasoning steps can be performed by external modules, which may improve generalization. However, this would require additional compute to load the additional models and introduce more system-building complexity. Likewise, we find that current open-source VLMs are unable to generate the bulk of the reasoning. While proprietary LLMs like ChatGPT may be able to do so, they are naturally not trained to automatically generate reasonings of the expected format, and thus would require in-context examples or long prompt instructions on formatting (like with Fig. 10), slowing inference speeds. Furthermore, API calls would need to be made for each new image, incurring costs.

Another possibility would be to use external modules to give feedback on generated reasoning chains. In practice, this is difficult to do zero-shot with off-the-shelf pre-trained models; most works using VLMs as feedback for learning or inference typically need to fine-tune them for their specific use case [75, 76].

Annotate the training trajectory with reasoning

## Specification of the experimental setup

You're an expert reinforcement learning researcher. You've trained an optimal policy for controlling a robotic arm. The robot successfully completed a task specified by the instruction: "unfold the cloth from top right to bottom left". For that purpose, the robotic arm executed a sequence of actions. Consecutive states that were visited can be characterized by the following features:

```python
trajectory_features = {
    0: "stop"
    1: "stop"
    2: "move forward left"
    3: "move forward down"
    4: "move forward down"
    ...
    36: "stop"
}
```

Each entry in that dictionary corresponds to a single step on the trajectory and describes the move that is about to be executed.

## Scene description

The robot is operating in the following environment. A black and red toy stove with a yellow banana in a silver pot, a blue toy brush, and a purple towel on the counter, surrounded by white tiled walls and a grey sink.

## Your objective

I want you to annotate the given trajectory with reasoning. That is, for each step, I need to know not only which action should be chosen, but importantly what reasoning justifies that action choice. I want you to be descriptive and include all the relevant information available. The reasoning should include the task to complete, the remaining high-level steps, the high-level movements that should be executed and why they are required, the premises that allow inferring the direction of each move, including the locations of relevant objects, possible obstacles or difficulties to avoid, and any other relevant justification.

### Begin by describing the task

Start by giving an overview of the task. Make it more comprehensive than the simple instruction. Include the activity, the objects the robotic arm interacts with, and their relative locations in the environment. Then, describe the high-level movements that were most likely executed, based on the task that was completed and the primitive movements that were executed. Then, for each high-level movement write the interval of steps that movement consists of. Also, for each high-level movement write a justification for why it should be executed. Write an answer for this part using markdown and natural language. Be descriptive and highlight all the relevant details, but ensure that your description is consistent with the trajectory that was executed, specified by the features listed above in the `trajectory_features` dictionary.

### List the reasonings for each step

Finally, for each step describe the reasoning that allows to determine the correct action. For each step describe the remaining part of the objective, the current progress, the objects that are still relevant for determining the plan, and the plan for the next steps, based on the available features. Start the reasoning from a high level and gradually add finer features. I need you to be descriptive and very precise. Ensure that the reasoning is consistent with the task and the executed trajectory. Write the answer for this part as a Python-executable dictionary. For every step in the initial trajectory there should be exactly one separate item of the form <step id>:<reasoning>. Do not group the answers. The final dictionary should have exactly the same set of integer keys as the dictionary of features provided in the `trajectory_features` dictionary above. The reasoning should be a single string that describes the reasoning in natural language and includes all the required features.

Each reasoning string should have the following form:
- Describe the full task that remains to be completed (but only describe what remains), and place it inside a tag <task>.
- Describe the complete high-level plan for completing the remaining task (the list of remaining high-level steps), and place it inside a tag <plan>.
- Describe the high-level step that should be executed now (chosen from the list of high-level steps), and place it inside a tag <subtask>.
- Describe why the chosen high-level step should be executed now, which features of the current environment influence that decision, and how it should be done. Place it within a tag <subtask_reason>.
- Describe the current primitive movement of the arm that needs to be executed, and place it inside a tag <move>.
- Describe why the chosen movement should be executed now and which features of the current environment influence that decision. Place it inside a tag <move_reason>.

## Task summary

Here is a breakdown of what needs to be done:

- Describe the task.
- Describe the high-level movements that were executed, based on the completed task and the listed features.
- Describe the plan for the solution that allowed the robot to complete the task successfully.
- For each step on the trajectory, describe the reasoning that leads to determining the correct action. The reasoning should be descriptive and precise. You should provide exactly one reasoning string for each step on the trajectory specified by `trajectory_features`.
- At the very end of the response, write a single label FINISHED to indicate that the answer is complete.

Figure 10: Prompt used for Gemini to generate plans, subtasks, and movement labels.

```
# Objective

You're an expert reinforcement learning researcher.  You've trained a policy for controlling a robotic arm.  The policy
computes the correct action based on a reasoning that leads to it, which includes the task that remains to be completed,
the plan for completing that task, and the subtask that currently needs to be done.  I want you to prepare such a reasoning,
based on a feedback from a user of that robot.

The reasoning must have the following elements:
- TASK: the task that remains to be done.
- PLAN: a list of high-level steps that need to be executed.
- SUBTASK REASONING: reasoning that determines the current subtask.
- SUBTASK REASONING: reasoning that determines the current subtask.
- SUBTASK: the current subtask that should be executed.
- MOVE REASONING: reasoning that determines the current move.
- MOVE: the current move that should be executed

Write the answer as a python string.  It will be used as an additional input for the policy, so keep the format exactly as
described.

# Examples

Given the task "Put the tomato inside the pot on the left burner" and feedback "you are too low, move up", the reasoning
should be "TASK: Put the tomato inside the pot on the left burner.  PLAN: Go to the tomato, grasp it, transport it to the
stove, position it in the pot.  SUBTASK REASONING: The tomato is grasped.  The tomato is already near the pot, but below its
edge.  SUBTASK: Position the tomato in the pot.  MOVE REASONING: The pot is above current position.  Move the arm up.  MOVE:
Move up."

Given the task "place the silver lid on the silver pot on the upper right of the table" and feedback "move to the pot", the
reasoning should be "TASK: The lid needs to be placed on a silver pot on the upper right part of the scene.  PLAN: First
move to the lid, then grip it, then move to the pot, then place the lid on the pot.  SUBTASK REASONING: The lid is gripped,
so it should be moved to the pot.  SUBTASK: Move to the pot."

Given the task "move the fork to the bottom left side of the counter" and feedback "move down to grasp the fork", the
reasoning should be "TASK: Pick up the fork and move it to the bottom left side of the counter.  PLAN: 1.  Move to the fork.
2.  Pick up the fork.  3.  Move to the bottom left side of the counter.  4.  Put down the fork.  SUBTASK REASONING: The fork
is the first object that needs to be reached.  SUBTASK: 1.  Move to the fork.  MOVE REASONING: The fork is still downward
from the current position of the arm, so the arm continues to move that direction.  MOVE: move down"

Given the task "remove the cylinder from the green cube and place it on top of the red cube" and feedback "close", the
reasoning should be "TASK: Remove the cylinder from the green cube and place it on top of the red cube.  PLAN: Approach the
green cube, close the gripper around the cylinder, move the cylinder towards the red cube, open the gripper to place the
cylinder on top of the red cube.  SUBTASK REASONING: The arm is now in contact with the cylinder, so it should close the
gripper to grab it.  SUBTASK: Close the gripper MOVE REASONING: The cylinder has already been reached and the gripper is
closing.  MOVE: Close gripper"

Given the task "pick up the towel" and feedback "go right", the reasoning should be "TASK: Pick up the towel.  PLAN: Move to
the towel, Grasp the towel, Pick up the towel SUBTASK REASONING: The arm should reach the towel first.  SUBTASK: Move to the
towel.  MOVE REASONING: The towel is to the right, so the arm should move right.  MOVE: move right"

# The current task

The policy generated the following reasoning:  "TASK: Put the mushroom in the white cup.  PLAN: Move to the mushroom, pick
it up, move to the cup, put the mushroom in the cup.  SUBTASK REASONING: The mushroom is picked up, and the cup is the next
object to interact with.  SUBTASK: Move to the cup.  MOVE REASONING: The cup is positioned below the arm.  MOVE: Move down.
GRIPPER POSITION: [111, 61] VISIBLE OBJECTS: a white cup [124, 25, 176, 113], a wooden table [13, 21, 241, 248], a wooden
table [10, 21, 249, 250]"

Given the task "put the mushroom in the white cup" and feedback "no, the cup is actually in front of the gripper", what
should be the reasoning?
```

Figure 11: Prompt used for ChatGPT during human intervention experiments

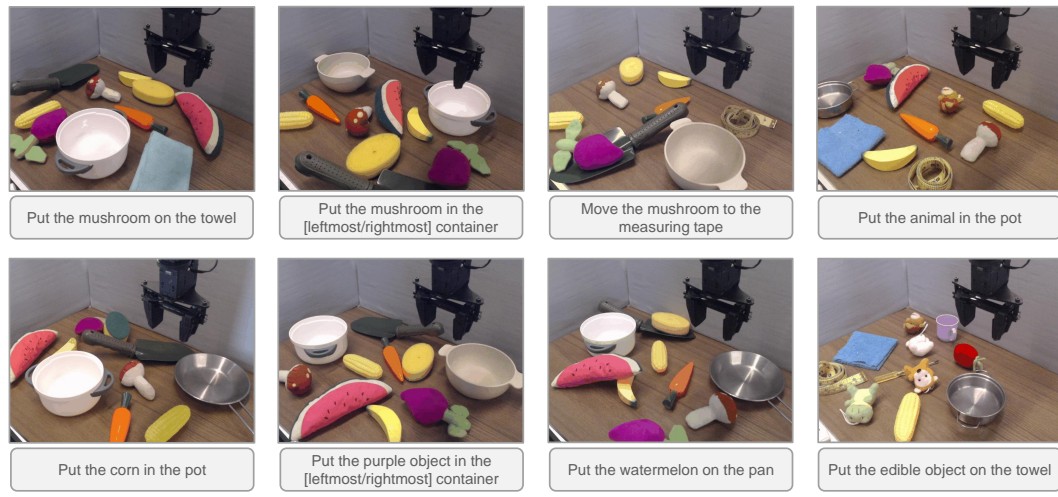

| | |
|---|---|
| Put the mushroom on the towel | Put the mushroom in the [leftmost/rightmost] container |
| Move the mushroom to the measuring tape | Put the animal in the pot |
| Put the corn in the pot | Put the purple object in the [leftmost/rightmost] container |
| Put the watermelon on the pan | Put the edible object on the towel |

Figure 12: Example starting scenes for each of the cluttered evaluation tasks.

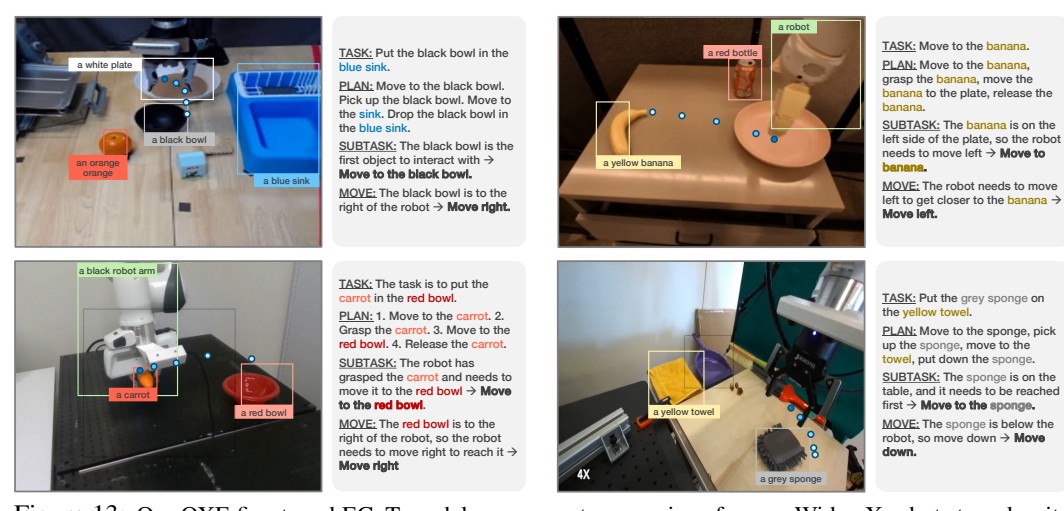

Figure 13: Our OXE fine-tuned ECoT model can generate reasonings for non-WidowX robots too, despite never having seen reasoning annotations for said embodiment.

