# OpenReview forum: "Robotic Control via Embodied Chain-of-Thought Reasoning"
_robot-learning.org/CoRL/2024/Conference — CoRL 2024_

### Official Review · Reviewer_Nt3h · 2024-07-20
**The review for Embodied CoT**

**Originality:** 4
**Technical Quality:** 5
**Clarity Of Presentation:** 5
**Potential Impact:** 4
**Recommendation:** 4
**Confidence:** 4

**Review:**

**Summary**:
The paper proposes a novel robotic control pipeline by including chain-of-thought in the end2end VLA backbone.
The chain-of-thought involves natrual language subtask, next step motion planning, subtask reasoning, visible object localization, boudning box prediction and other combination. The model is build upon OpenVLA, and the experiments over the self-collected dataset demonstrates its efficiency.

**Strength**:
1. The novel framework for predicting mid-level results like the language subtasks and the bounding box are reasonable. Adding these as pre-output of the VLA model is an approprate pattern.

2. The chain-of-thought is very diverse, which includes natrual language subtask, next step motion planning, subtask reasoning, visible object localization, boudning box prediction. These mainly contains all the essential components i could image.

3. The collection of the training dataset is very reasonable and with high quality.

**Weakness**:
1. Some experiments on simulator could be helpful for fair comparison with existing works.

2.More ablation on key components of the CoT and different combination should be added.

**Quality Of The Limitations Section:**

2

**Questions For Rebuttal:**

See weakness.

**Robotics Focus:**

4

**Summary Of Paper:**

The paper proposes a novel robotic control pipeline by including chain-of-thought in the end2end VLA backbone. The CoT involves language subtask, boudning box and other combination, and the experimental results is promising.

**Summary Of Recommendation:**

The paper proposes a novel robotic control pipeline by including chain-of-thought in the end2end VLA backbone. The CoT involves language subtask, boudning box and other combination, and the experimental results is promising.

---

### Official Review · Reviewer_LC5L · 2024-07-21
**CoT being used for vision-language-action models**

**Originality:** 4
**Technical Quality:** 4
**Clarity Of Presentation:** 4
**Potential Impact:** 4
**Recommendation:** 4
**Confidence:** 4

**Review:**

The key idea of this paper, i.e., applying the chain of thought technique to vision-language-action models, is solid and well presented. The reviewer is not aware of existing references studying this idea (the other reviewers can correct me if I am wrong). The baselines were well selected and represent the state of the art, which is good too.

The "chain" of thought in the proposed approach was manually created and predefined, as demonstrated in Figure 4. While it worked for the tasks in the dataset used for evaluations in this research, it's a limitation and dynamically figuring out the reasoning chain can potentially be an improvement. It's unclear how sensitive the performance is to the current pipeline. For instance, if the bounding boxes are replaced with 3D poses, it's unclear how the results would be different.

This is a minor comment. The interactive correction part doesn't connect to this work well. It's pointing to a different direction (HRI) while the focus of this paper is a novel prompting strategy for VLAs.

**Quality Of The Limitations Section:**

3

**Questions For Rebuttal:**

Was the collected dataset used for fine tuning the VLAs? What if we use the training data as examples for few-shot inference?

Are there robot tasks that the current pipeline is inapplicable to?

How is the performance sensitive to the current pipeline? For instance, what if we flip the order of the first two steps? What if the 2D bounding boxes are replaced with 3D boxes?

**Robotics Focus:**

4

**Summary Of Paper:**

This paper looked into a pretty straightforward (yet not investigated) idea, which is on applying the chain of thought (CoT) approach to vision-language-action models (VLAs). It has been evident that the CoT technique is useful in large language models (LLMs) for different types of reasoning tasks. Recent advances on VLAs, such as the series of RT research, shows that pretrained foundation models are able to directly  perform robot control. The proposed approach is called ECoT (embodied chain of thought reasoning) for computing VLA policies for robots. OpenVLA and RT-2-X were used as baselines for comparisons. Results showed that ECoT performed better than the two baselines on real robots and human feedback could further improve the performance.

**Summary Of Recommendation:**

Nice paper that described a novel idea that can potentially produce good impact to the community. The experiments were well performed and the results were convincing.

---

### Official Review · Reviewer_DLrg · 2024-07-23
**Interesting approach for incorporating embodied chain of thought reasoning to OpenVLA. The work compared to strong baselines and demonstrates improved OOD performance on a selection of tasks. More details on the training and inference pipeline needed and further clarification questions on results, additional baselines, incorporating feedback and inference speed are desired.**

**Originality:** 4
**Technical Quality:** 3
**Clarity Of Presentation:** 3
**Potential Impact:** 3
**Recommendation:** 4
**Confidence:** 4

**Review:**

Strengths:

Important extension to prior work – training VLAs to output step-by-step (CoT) grounded reasoning steps

A pipeline for generating the grounded CoT training data at scale

The proposed approach is compared to a strong set of baselines (OpenVLA and RT-2-X) as well as an important ablation (Naïve CoT) is considered that highlights the need for embodied reasoning in robotics tasks

ECoT policy outperforms OpenVLA and RT-2-X  on most generalization tasks for in-distribution viewpoints.

Weaknesses:

Augmentation to the Bridge v2 dataset seems tedious and unreliable. High reliance on pre-trained models and no measures are taken for ensuring quality/correctness of the extracted features.

More details on the training objective and prediction pipeline could improve the clarity of the paper.

Results are demonstrated on a very small set of  tasks. How does the policy perform in clutter when the object detection modules will likely find it very challenging to identify objects. It would be useful to show results (even in simulation) on a set of more challenging tasks especially in clutter

If the human is not in the loop, the policy is mostly open-loop. No replanning framework considered for when object-detection fails, sub-tasks are incorrect or policy fails? One of the benefits of CoT can be to reason about past mistakes and replan accordingly. However, since the training data does not contain any failure and recovery scenarios, the current CoT framework cannot do that.

Learned policy is very sensitive to camera viewpoints

Even though steps are taken to improve inference time, speed of execution is still limiting

Clarification questions:

What are the exact inputs and outputs of the policy? What is the training objective? Is the model predicting everything – task, plan, subtask, bounding boxes, gripper position and move primitives? Are the predictions sequential? If so, is the prediction pipeline similar to the data generation pipeline, i.e. first a task description is generated, then a scene description, then the bounding boxes (given the task and scene), then the gripper position and finally the plans, subtasks and move primitives (given the task, scene, bounding boxes and gripper position).

Instead of learning to predict the bounding boxes, why not, during inference, utilize Prismatic VLM and Grounding Dino to get the bounding boxes and gripper position, while using the rest of the prediction pipeline to get the embodied CoT? This could improve generalization to OOD objects.

RT-2-X seems to have a 0% success rate for the “Put mushroom in tall cup” while OpenVLA significantly outperforms ECoT. This is very surprising since RT-2-X is trained on much more data. Very curious to know the failure modes for this task for RT-2-X and ECoT.

The OOD Instructions seem quite challenging since most object detection modules will really suffer on ‘not <object>’ predictions. It is intriguing to see that both RT-2-X and ECoT perform better with Out-of-Distribution Viewpoint than with In-Distribution Viewpoint. It will be useful for the authors to share the reasoning behind that.

In section 5.4, it is unclear why ECoT benefits significantly more than the baselines. In the example shown, it seems like the human intervenes and corrects the object detection. This should benefit all the baselines equally.

For efficient Chain-of-Thought Inference, did the authors consider updating the high level plan based on feedback from the environment for instance based on a success/failure detection module (once a subtask is completed)?

In section 5.4, ChatGPT is used for correcting the reasoning chain. It is evident that including task information in the prompt allows ChatGPT to generate grounded CoT reasoning steps. Curious to know the author's thoughts on another baseline where OpenVLA is trained to perform language-conditioned  subtasks and during inference the subtasks are generated from ChatGPT using grounded CoT reasoning as is done in section 5.4.

The failure cases mentioned in the paper and in the appendix seem like object detection or scene understanding failures. It would be interesting to see if failure modes where the object detections or scene understanding are correct but the method fails to reason about them using ECoT.

As an important implementation detail, it would be useful to the reader to know more about how, in section 5.5, part of the model prediction is kept fixed while the rest is updated? In section 5.4, it is mentioned that “we feed this corrected chain back into our policy and continue execution”, how is this done exactly? Does the policy take as input an initial reasoning chain? What are the exact inputs and outputs of the policy?

**Quality Of The Limitations Section:**

2

**Questions For Rebuttal:**

More details on the training objective and prediction pipeline including some of the clarification questions mentioned above could improve the clarity of the paper.

To evaluate if the accelerated CoT inference approaches mentioned in the paper are sufficient, the authors should consider providing comparisons of inference times against the baselines considered.

It would be useful to include the set of objects considered for in-distribution and OOD tasks in the appendix.

**Robotics Focus:**

4

**Summary Of Paper:**

This work builds upon OpenVLA and incorporates chain-of-thought reasoning while grounding it in sensory observations and the robot state. Since OpenVLA builds on relatively small open-source VLMs, its chain-of-thought reasoning capability does not match that of large closed LLM models. Furthermore, this step-by-step reasoning should be grounded in the robotics task. Thus, the authors propose training a VLA policy (using the OpenVLA architecture) to output CoTs and actions given input instructions and observations, ensuring that  both are firmly grounded in the agent’s environment.

**Summary Of Recommendation:**

Interesting approach for incorporating embodied chain of thought reasoning to OpenVLA. It is a useful contribution towards improving the capabilities of VLA models.

---

### Author Rebuttal · Authors · 2024-08-07

We provide a revised version of the paper attached with all discussed edits and additions in the attached zip file.

---

### Decision · Program_Chairs · 2024-09-04

**Decision:**

Accept

**Comment:**

### Strengths

- Re: Novelty: All reviewers agree that the paper should be considered a significant advance, as it introduces a novel robotic control pipeline that integrates chain-of-thought (CoT) reasoning into the end-to-end vision-language-action (VLA) model. R-Nt3h mentions that the CoT framework includes diverse components such as natural language subtasks, motion planning, and object localization, which they further suggest are appropriate and comprehensive for the task.
- Re: Evaluation: R-DLrg highlights that the approach is compared against strong and relevant baselines, and includes important ablations (e.g., Naïve CoT) that highlight the benefits of embodied reasoning. R-DLrg and R-LC5L note that the proposed ECoT approach outperforms existing models (OpenVLA and RT-2-X) in most generalization tasks and real robot evaluations. R-Nt3h suggests that the training dataset used is of high quality and is well-collected.
- Re: Clarity: R-LC5L and R-Nt3h suggest that the paper is generally well-organized, clear, and the experiments are well-performed, making the results convincing.

### Weaknesses

- Re: Method: R-DLrg cites dataset augmentation issues, wherein the augmentation of the Bridge v2 dataset is seen as tedious and unreliable, with high dependence on pre-trained models and no measures for quality assurance. R-LC5L sees the interactive correction mechanism as somewhat disconnected from the main focus of the paper, which they assert is the CoT strategy. R-DLrg expressed concern over the fact that the current policy is mostly open-loop, with no replanning framework for handling failures or incorrect sub-tasks — limiting the policy’s ability to adapt and recover from mistakes. R-DLrg further suggests that the learned policy is very sensitive to camera viewpoints, which may affect its generalizability, and that the speed of execution remains a limiting factor, despite efforts to improve inference time.
- Re: Evaluation: R-Nt3h points out that the absence of experiments in a simulator for fair comparison with existing works is noted as a limitation. R-DLrg expressed concern over a lack of challenging tasks, asserting that the experimental results are limited to a small set of tasks and that the paper does not address performance in more complex scenarios such as cluttered environments. R-Nt3h calls for more ablation studies on key components of the CoT and their combinations to better understand their impact.
- Re: Clarity: R-LC5L expressed concern over the lack of clarity on how sensitive the performance is to different pipeline components, such as the impact of replacing 2D bounding boxes with 3D boxes.

### Post-rebuttal Meta Review Statement

I applaud the reviewers and authors for engaging in a productive discussion that resulted in several improvements to the paper, especially as it relates to the addition of new experimental evaluations (in response to R-Nt3h and R-DLrg, enabling the latter to raise their score) and sensitivity analysis of the policy to alternative reasoning steps and orderings (in response to R-LC5L).

I recommend for Accept as Oral, given the initial reviews and the extent to which the reviewer feedback was satisfied.

Still, I encourage the authors to follow through with all the promised changes and to incorporate all additional reviewer requests, to improve the paper even further.